# A Marked Low-Grade Inflammation and a Significant Deterioration in Metabolic Status in First-Episode Schizophrenia: A Five-Year Follow-Up Study

**DOI:** 10.3390/metabo12100983

**Published:** 2022-10-17

**Authors:** Madis Parksepp, Liina Haring, Kalle Kilk, Egon Taalberg, Raul Kangro, Mihkel Zilmer, Eero Vasar

**Affiliations:** 1Institute of Clinical Medicine, University of Tartu, 50417 Tartu, Estonia; 2Psychiatry Clinic of Viljandi Hospital, 71024 Viljandi, Estonia; 3Psychiatry Clinic of Tartu University Hospital, 50417 Tartu, Estonia; 4Centre of Excellence for Genomics and Translational Medicine, Institute of Biomedicine and Translational Medicine, Univesignallingrsity of Tartu, 50090 Tartu, Estonia; 5Institute of Mathematics and Statistics, University of Tartu, 51009 Tartu, Estonia

**Keywords:** first-episode psychosis, schizophrenia spectrum disorder, antipsychotic treatment, cytokines, acylcarnitines, beta-oxidation, low-grade inflammation, metabolic syndrome, obesity

## Abstract

The objective of this study was to evaluate how schizophrenia spectrum disorders and applied long-term (5.1 years) antipsychotic (AP) treatment affect the serum level of acylcarnitines (ACs), cytokines and metabolic biomarkers and to characterize the dynamics of inflammatory and metabolic changes in the early course of the disorder. A total of 112 adults participated in the study (54 patients with first-episode psychosis (FEP) and 58 control subjects). Biomolecule profiles were measured at the onset of first-episode psychosis and 0.6 years and 5.1 years after the initiation of APs. The results of the present study confirmed that specific metabolic–inflammatory imbalance characterizes AP-naïve patients. Short-term (0.6-years) AP treatment has a favourable effect on psychotic symptoms, as well as the recovery of metabolic flexibility and resolution of low-level inflammation. However, 5.1 years of AP treatment resulted in weight gain and increased serum levels of interleukin (IL)-2, IL-4, IL-6, IL-10, interferon-γ, hexoses, acetylcarnitine, short-chain ACs (C3, C4) and long-chain ACs (C16:2, C18:1, C18:2). In conclusion, despite the improvement in psychotic symptoms, 5.1 years of AP treatment was accompanied by a pronounced metabolic–inflammatory imbalance, which was confirmed by the presence of enhanced pro-inflammatory activity and increased obesity with changes in the metabolism of carbohydrates, lipids, and their metabolites.

## 1. Introduction

Pathophysiologic theories and the pharmacotherapy of schizophrenia spectrum disorders (SSD) are traditionally related to monoamine neurotransmitters. However, heterogeneity in clinical manifestations, illness course and the benefits of adverse effects of antipsychotic (AP) treatment suggest that additional mechanisms are also involved. A plethora of the scientific literature demonstrates that low-grade inflammation, disturbed immunomodulatory functions and changes in lipid composition might be associated with the onset and progression of SSD [1,2,3,4]. Although there is compelling evidence that changes in the blood levels of immune and metabolic biomarkers occur during the first episode of psychosis (FEP), further research is needed to reliably elucidate the molecular mechanisms involved in the disease and treatment.

The ability to switch between available energy substrates is decisive for efficient cell and whole-organism functioning. A metabolically flexible state exists when there is a quick shift between glucose and fatty acids (FAs) during the transition between the fed and fasting state and impaired flexibility is involved in the development of metabolic disorders [5].

In line with previously published meta-analyses [2,4,6], research performed by our group [7,8] established the significantly higher levels of anti-, and pro-inflammatory cytokines (interleukin (IL)-4, IL-6) in drug-naïve FEP patients compared to control subjects (CSs), and AP treatment was related to decreased concentrations of cytokines (IL-2, IL-4, IL-6, IL-8, interferon (IFN)-γ). In addition, the significantly elevated ferritin levels in FEP indicate cell damage, favorably responding to AP treatment [8]. Furthermore, our previous research [9] suggests that the presence of chronic disease is associated with a significant increase in cytokines, intensified oxidative stress, and a significantly increased number of individuals with prediabetes or diabetes compared to FEP. Although APs alleviate the health risks arising from low-grade inflammation in the body during the FEP onset, these drugs often cause serious weight gain and increase the risk of cardio-metabolic illnesses. Accumulating evidence indicates that at least one group of patients with SSD presents a higher prevalence of metabolic syndrome (MetS) together with its related abnormalities, including glucose and lipid metabolism disturbances [10,11]. Additionally, weight gain is closely related to adipocyte dysfunction, which is manifested by increased levels of pro-inflammatory adipokines [12], including leptin [13]; high circulating concentrations of these substances are present in patients with SSD, during both the early and later phases.

It has been demonstrated that the acylcarnitine (AC) profile, or the concentration of ACs (esters arising from the conjugation of FAs with L-carnitine) with specific chain lengths could characterize the intracellular energy metabolism pattern and could be used as a marker of metabolic dysfunction [14,15]. The body carnitine pool is comprised of L-carnitine (C0), acetylcarnitine (C2), short-chain (C3–C5), medium-chain (C6–C12), and long-chain (C14–C20) ACs [16]. In contrast with short-chain ACs (SCACs), which are produced from glucose, amino acids and FA degradation, medium- and long-chain (LC) ACs (LCACs) are mainly synthesized and metabolized in mitochondria. LCFA are actively transported across the cell membrane. In the cytoplasm, FAs can either be stored in lipid droplets or transformed into LC acyl-CoA. The carnitine palmitoyltransferase (CPT) system then transports LC acyl-CoA from the cytoplasm into the mitochondrial matrix for β-oxidation. CPT1 converts LC acyl-CoAs into LCACs. Carnitine translocase (CACT) moves AC across the inner mitochondrial membrane as carnitine is exported. Inside the mitochondria, LCAC is converted back into LC acyl-CoAs for β-oxidation by CPT2, which can then be further oxidized via the tricarboxylic acid (TCA) cycle and respiratory chain to provide energy [15,17]. CPT1 deficiency can be screened using the ratio of (C16 + C18:1)/C0 and CPT2 deficiency can be screened using the ratio of (C16 + C18:1)/C2. Deficiencies in these enzymes or impaired functions may cause incomplete mitochondrial FA β-oxidation, resulting in the accumulation of ACs [18]. The most widely distributed LCFAs are the 16- and 18-carbon FAs. Besides being a well-known marker of incomplete FA oxidation and mitochondrial metabolism, studies highlight the multifunctional role that these LCACs play in the cell. These lipids have the ability to influence cell signalling cascades, and thereby induce insulin resistance [17] or activate classical pro-inflammatory signalling pathways [19].

Among the numerous factors potentially implicated in the regulation of the CPT genes, including eicosanoids, saturated and unsaturated LCFAs as well as synthetic ligands (i.e., fibrates), special consideration has been paid to peroxisome proliferator-activated receptors (PPARs), since they are crucial transcriptional factors involved in FA oxidation, as well as glucose and lipid metabolism throughout the body [20]. PPARs belong to the nuclear receptor superfamily of ligand-activated transcription factors and, to date, three PPAR isoforms (PPARα, PPARβ/δ, and PPARγ) have been identified. These isotypes have complementary and distinct metabolic activities, depending on their tissue distribution and their specific ligands (reviewed in Lamichane et al. [20]). Additionally, PPARs have been also recognized as playing an important role in the control of various types of the inflammatory response (see Toobian, et al., for review [21]). In brief, PPARα is mainly expressed in the liver, adipose tissue, immune and other cells that regulate lipid metabolism. During a fed state, PPARα controls de novo lipogenesis to provide lipids for storage. However, during fasting, PPARα activity shifts to FA uptake and fatty acid β-oxidation [22]. PPARγ is mainly expressed in adipocytes or liver, where it regulates the expression of genes that are involved in lipid storage and adipocytes differentiation [23]. In parallel, PPARγ is abundantly expressed in macrophages, dendritic cells, as well as in B and T cells, where it regulates macrophage activation and the repression of inflammation [24]. Additionally, the gamma subtype also presents in pancreatic β-cells, and activation of PPAR-γ causes insulin sensitization and enhances glucose metabolism [25]. PPARδ/β is involved in the modulation of macrophage-derived inflammation and FA uptake, transport, and β-oxidation as well as insulin secretion and sensitivity [26]. The mechanism of PPARs anti-inflammatory properties is very complex and takes different forms. It has been demonstrated that PPARs can directly extenuate the expression of inflammatory responses genes, and since many anti-inflammatory genes are regulated by nuclear factor kappa-B (NF-*κ*B) pathways, the activation of all PPARs represses NF-*κ*B signalling, which decreases the inflammatory cytokine production by different cell types [27].

Although the role of the complex interaction of metabolic status changes and low-grade inflammation is emphasized in SSD, studies often focus on one side. Moreover, there is a need for longitudinal studies to increase the understanding of changes detected at specific timepoints in the levels of substrates measured in the blood serum of SSD patients. In addition, it should be taken into account that the course of the disease over time is dynamic, and the substrate profile is affected by short- and long-term AP treatment. Therefore, to understand the inter-relationships of levels of inflammatory and metabolic status in SSD patients, we designed a longitudinal research project in a naturalistic setting. We included AP-treatment-naïve FEP patients who were followed up for an average of 5 years and control subjects (CSs) in the study. We collected data on their body mass index (BMI), and simultaneously assessed the levels of anti- and pro-inflammatory markers and substrates reflecting metabolism (including adipokines and short- and long-chain ACs). The study aimed to comprehensively describe the changes in measured biomarker profiles occurring at different timepoints of the disease, against the background of AP treatment, to better understand the underlying mechanisms of low-grade inflammation and altered whole-body metabolism associated with SSD.

## 2. Materials and Methods

### 2.1. Participants

A total of 112 adults participated in the study. Patients with FEP (*n* = 54, 59% men) were recruited at the time of their first clinical contact for psychotic symptoms at the Psychiatric Clinic of Tartu University Hospital, Estonia. The inclusion criteria were as follows: patients with FEP, the duration of the untreated psychosis less than 3 years, no AP use before the study, and male or female participants between 18 and 45 years old. If necessary, patients received benzodiazepines the night before the first blood collection. The administration of sedative drugs is often standard practice, as the acute psychotic state is accompanied by agitation, and motor restlessness and patients often have pronounced sleep disturbances, which need to be alleviated with fast-acting psychotropic drugs. The exclusions were as follows: patients who had organic or drug-induced psychosis or psychotic disorders due to other medical conditions. FEP diagnoses were based on clinical interviews according to the International Classification of Diseases, Tenth Edition (ICD-10) [28] criteria, and approved by two clinical psychiatrists. Patients’ diagnoses were F23.0 (*n* = 9), F23.1 (*n* = 11), F23.2 (*n* = 15), F23.3 (*n* = 2), F21 (*n* = 1), F20.09 (*n* = 13), and F20.39 (*n* = 3) at baseline. After the recruitment, all FEP patients received AP medication. The history of used APs was collected according to reviews of patients’ medical charts. No restrictions were made in terms of usage of specific pharmacological substances due to a naturalistic and longitudinal study design. During the study, patients were treated with various doses and types of APs. At the time of the follow-up, the daily doses of AP drugs in each prescription were converted into AP loading using the prescribed daily dose and the mean theoretical chlorpromazine equivalent (CPZE) doses according to Gardner et al. [29] were calculated. CPZE dose is defined as the dose of AP drug, which is equivalent to 100 mg of oral chlorpromazine. For patients prescribed more than one APs, the daily dose was calculated by summing the prescribed CPZE doses of each medication. Moreover, mood stabilizers, antidepressants, or hypnotics were used according to clinically relevant circumstances. Patients were examined prospectively. At an average of 0.6-year follow-up, the patient sample consisted of 47 patients (51% men), and at an average of 5.1-year follow-up, the sample comprised 38 patients (43% men). During the monitoring period, the patient dropout rate for different reasons was 30%. The main causes of discontinuation were related to their decision to stop AP treatment or because they had changed their place of residence. The patients’ diagnoses at the second follow-up were F20.0 (*n* = 28), F20.1 (*n* = 1), and F25 (*n* = 9). The control group (CSs) consisted of 58 subjects who were recruited through advertisements. Of these mentally healthy participants, 44% were male. As it was a naturalistic study, substance abuse was not an exclusion criterion for either group. Twenty-one patients (39%) had smoked cannabis before the FEP. Nineteen patients (35%) reported rare cannabis consumption during the five-year monitoring period, and three of them (men) met the criteria for cannabis use disorder. Fifteen CSs (26%) had tried cannabis at least once during their lifetime. None of them met the criteria of cannabis use disorder. All participants had no history of chronic medical illness, including metabolic disorder and no indications of acute infectious disease, allergy, another immune-related disease or recently received vaccines at study entry. According to the study inclusion protocol, the presence of such problems would have excluded the person from inclusion in the study sample. During the follow-up visits, none of the patients included in the study had any immune system disorders that could cause abnormally low activity or oveactivity of the immune system. None of the study participants regularly used anti-inflammatory drugs during or before the study visits, e.g., to reduce fever or relieve pain associated with any physical illness. The same FEP and CSs groups were partially characterized in our previous studies [7,8,30], wherein we focused on describing the inflammatory biomarkers and metabolomic profiles in AP-naïve FEPs (*n* = 38) compared to controls (*n* = 37) and demonstrated changes in biomolecule profiles over 0.6 years. To describe even longer-term changes due to disease and treatment, we then increased the total number of subjects in the study, also collected patient data at 5.1 years after initiation of AP therapy, and described changes in amino acid, biogenic amine profiles and endocannabinoids in the FEP group, before initiating AP therapy, as well as over 0.6- and 5.1-year periods [31,32]. At present, 5.1-year longitudinal data from 54 patients and data from 58 controls (partially the same cohort as described previously) were included in this study.

### 2.2. Procedure

Fasting serum samples, clinical data, BMI and waist circumference of the patients with a first psychotic disorder were assessed at three consecutive timepoints: on admission, at the first follow-up (the average time between visits was 0.6 ± 0.06 years), and at the second follow-up (the average time between the first and the third visits mean duration 5.15 ± 1.25 years). Serum sample collection from CSs and patients started in June 2009, and the last participant was recruited in November 2014. The 5-year follow-up serum sample gathering began in May 2013 and ended in November 2017, and only comprised the patients’ group. Blood samples were collected using the standard antecubital venipuncture technique between 09:00 and 11:00 a.m. Blood (5 mL) was sampled in anticoagulant-free tubes and kept for 1 h at 4 °C (for platelet activation). The blood was subsequently centrifuged at 2000× *g* for 15 min at 4 °C, and the serum was aspirated, divided into aliquots, and immediately frozen and stored at −20 °C for up to 2 weeks or at −80 °C for longer periods.

We used the Brief Psychiatric Rating Scale (BPRS) [33] to assess the presence of psychopathological symptoms in patients. The BPRS consists of 18 symptoms and each item was measured on a seven-point Likert scale from “not present” to “extremely severe”. The total score of the scale was used to describe the severity of the patient’s psychopathology. Data from CSs were collected cross-sectionally. Study data were collected and managed using REDCap electronic data capture tools hosted at the University of Tartu, Estonia [34,35].

### 2.3. Measurement of Biomarkers

#### 2.3.1. Measurement of Cytokines

We applied a high-sensitive biochip array technology (Randox Biochip, RANDOX Laboratories, Crumlin, U.K., Cytokine and Growth Array for Evidence Investigator) to measure the profile of circulating cytokines. The following cytokines were assessed according to the manufacturer’s protocol: IL-2, IL-4, IL-6, IL-8, IL-10, IFN-γ, IL-1A, IL-1B, TNF-α, and MCP-1. A more detailed description of these measured markers in relation to the partially same cohort has been provided elsewhere [7].

#### 2.3.2. Measurement of Metabolic Syndrome Biomarkers

The following inflammatory and metabolic biomarkers were measured by a biochip array technology (Randox Biochip, RANDOX Laboratories, Crumlin, U.K., Metabolic Syndrome Array I for Evidence Investigator): insulin, leptin, resistin, ferritin, and PAI-1. Relevant information about the analyte detection and quality-control method has been given previously [8].

#### 2.3.3. Measurement of Acylcarnitines

Serum level of ACs were determined with the AbsoluteIDQ p180 kit (BIOCRATES Life Sciences AG, Innsbruck, Austria) using the flow injection analysis tandem mass spectrometry ((FIA)−MS/MS), as well as the liquid chromatography ((LC)−MS/MS) technique. The Biocrates AbsoluteIDQ p180 kit is a commercially available targeted metabolomics assay, and is applied to many studies of human serum and plasma, including clinical ones. All measurements were performed as described in the manufacturer’s manual UM-P180. Identification and quantification of the metabolites was achieved using multiple reactions’ monitoring along with internal standards. From all statistically important changes in ACs in our study, we used only those values that were at least 2.3 times higher than the level of detection (LOD) given in the manual of the Biocrates AbsoluteIDQ p180 in the discussion. Details on the procedure and the sensitivity of the metabolite detection were described in our previous publication [30]. Calculation of metabolite concentrations was automatically performed by MetIDQ software (BIOCRATES Life Sciences AG, Innsbruck, Austria). To ensure data quality, they were checked based on the LOD. The Absolute IDQTM p180 kit allows for the simultaneous quantification of acylcarnitines (ACs) amongst other metabolites. Lipid side chain composition is described as Cx:y, where x denotes the number of carbons in the side chain and y denotes the number of double bonds. Average values of all measured biomarkers are presented in Appendix A.

### 2.4. Statistical Analyses

All data were checked for normality of distribution using the Shapiro–Wilk test. Normally distributed data (age, weight, BMI, and waist circumference) were analyzed using the Student’s *t*-test or repeated measures ANOVA, and mean differences were tested with the Scheffé post hoc test. Dichotomous data (gender and smoking status) were analyzed using the chi-square test. To examine the alterations in putative biomarker levels between CSs and FEP patients over time, linear mixed-effects (LME) models were used. Repeated measurements were handled by including a random intercept for participants in the model (since patients have different baseline values) and by allowing for time-dependent correlations between different measurements of each patient. The model intercept was set to represent putative biomarker levels of the control group and variables’ coefficients were compared to this intercept. Each set of analysis was adjusted for potential confounders: gender, age at the first visit, smoking status, and the time difference between the visits (time difference between expected time and given time). However, these covariates were not of primary interest in this study. Time dependence between measurements of each patient was modeled by continuous autoregressive correlation structures of order 1 and models were fitted using the maximum likelihood method. The candidate biomarker data were log-transformed before analysis to reduce the heterogeneity of variance. Moreover, several metabolite ratios were calculated as indicators of metabolic processes. We similarly treated longitudinal data in our previous studies. First, to identify dependent variables, which behave differently in the case of patients and the control group, we fitted two nested models to the data (i.e., reduced model with dependent variables with no patients’ specific independent variables compared to a more complex model with added terms allowing for the possibility for the value of the dependent variable to depend on patients’ type of visit and time between visits). We compared models using the likelihood ratio test, where the false discovery rate (FDR) procedure was implemented for multiple testing corrections [36], setting a cut-off of ≤5 × 10^−3^ [37]. Thereafter, the estimates from the LME analyses (fixed effects) were used to establish patients’ biomolecule profile alterations at three different timepoints. As we ran several LME models in parallel, and the selected metabolites belonged to classes of lipids that may share partially similar biosynthetic pathways and the correlation within the same lipid class is high, we considered the adjustment of the significance level to *p* ≤ 1 × 10^−4^ while selecting the metabolites with the most pronounced change, and the *p*-values between ≤5 × 10^−3^ and 1 × 10^−4^ were referred to as a trend change. The R [38] statistical language version 3.5.2 package nlme [39] and ANOVA-type diagnostic test were used to perform an analysis of the relationship between candidate biomarker levels differences among CSs and patients at three-time points, and Statistica software for Windows [40] was used for the other analyses. The visualization of error variances obtained from regression results was computed using R software package ggplot2 [41]. Residuals or errors terms were assumed to be independent among individuals but dependent within each participant.

## 3. Results

### 3.1. General Description of the Study Samples

The demographic and clinical characteristics of the study participants are presented in Table 1. In total, 112 participants were enrolled. There were no statistically significant differences between AP-naïve FEP patients and CSs in terms of age (*t*_(110)_ = 1.87, *p* = 0.07), gender (χ^2^_(1)_ = 3.58, *p* = 0.06), mean values of BMI (*t*_(110)_ = 0.34, *p* = 0.74) or waist circumference (*t*_(110)_ = −1.68, *p* = 0.09). In the patients’ group, AP treatment significantly reduced psychopathology (BPRS) score (*F*_(2,136)_ = 93.6, *p* < 1 × 10^−6^) but caused a significant increase in BMI and in waist circumference. Daily dose of APs, in terms of mean CPZE dose, did not differ between patients at 0.6-year of treatment (365 ± 163, range 60–680), and after 5.1 years of treatment (442 ± 297, range 75–1566; *F*_(1,81)_= 1.12, *p* = 0.29).

### 3.2. Changes in the Biomarkers, Body Mass Index and Waist Circumference of Patients with Schizophrenia Spectrum Disorder after 5.1 Years of Continuous Antipsychotic Treatment

Our primary analyses sought to uncover the robust effects of medication and illness on the measured parameters. Patient’s biomolecule data were compared to CSs over the 5.1-year trial period after adjusting for covariates (age, gender, smoking status, visit status, and the time difference between the visits). A set of two LME models was tested; both models used all the available data, but longitudinally collected patient-specific determinants were only taken into account in the unrestricted models. Details of the models used are given in Appendix A. According to FDR adjusted *p*-value (*p* ≤ 5 × 10^−3^) derived from model comparisons, unrestricted models provided a significantly better fit than the reduced model for the change in the levels of 40 (80%) biomolecules or calculated metabolite ratios, as well as BMI and waist circumference over time.

Thereafter, a series of LME regression models were conducted to test our main hypotheses. We previously reported the shifts in serum biomolecule levels in the partially same sample of the patients from the time of initiation of AP treatment up to 0.6 years [7,8,30]. Now, we focus on the changes that occurred after 5.1 years of AP treatment, but also consider the trends in the changes that we demonstrated earlier to highlight the dynamics over time.

#### 3.2.1. Body Mass Index and Waist Circumference Change during the 5.1 Years of Disease Duration and Antipsychotic Treatment

According to the results, BMI (*t*_(77)_ = 4.63, *p* < 1 × 10^−4^), and waist circumference (*t*_(77)_ = 5.50, *p* < 1 × 10^−4^) were strongly elevated for 5.1 years. Detailed results are provided in Appendix A and shown in Figure 1.

#### 3.2.2. Inflammatory and Metabolic Biomarker Changes among Patients with Schizophrenia Spectrum Disorder: At Antipsychotic-Naïve Status, after 0.6 Years and 5.1 Years of Continuous Antipsychotic Treatment

Consistent with our previous studies, serum levels of IL-2 (*t*_(73)_ = 4.92, *p* < 1 × 10^−4^), IL-4 (*t*_(72)_ = 5.95, *p* < 1 × 10^−4^), and IL-6 (*t*_(72)_ = 4.22, *p* = 1 × 10^−4^) were significantly elevated among the AP-naïve patients group and upward trends emerged for IL-1A (*t*_(73)_ = 3.44, *p* = 1 × 10^−3^), and for ferritin (*t*_(74)_ = 3.52, *p* = 7 × 10^−4^). These changes resolved with 0.6 years of AP treatment, except for IL-4, for which the upward trend was maintained (*t*_(72)_ = 3.43, *p* = 1 × 10^−3^).

By the fifth year, the positive effects of the seven months of AP treatment on the levels of inflammatory markers had disappeared, and a similar biomolecule profile emerged as in the pre-treatment state. Serum levels of IL-2 (*t*_(73)_ = 4.31, *p* = 1 × 10^−4^), IL-4 (*t*_(72)_ = 6.25, *p* < 1 × 10^−4^), IL-6 (*t*_(72)_ = 3.80, *p* = 3 × 10^−4^), IL-10 (*t*_(73)_ = 5.82, *p* < 1 × 10^−4^), IL-1A (*t*_(73)_ = 4.32, *p* < 1 × 10^−4^), and IFN-γ (*t*_(70)_ = 3.74, *p* = 4 × 10^−4^) were considerably elevated among the patient group. Additionally, due to the conservative statistical confidence level, there was an upward trend in the case of leptin (*t*_(74)_ = 3.43, *p* = 1 × 10^−3^) during the fifth year of the follow-up. Although the increase in ferritin levels at year five was not statistically confirmed, it should be considered a sign of worsening cellular damage. The test results are summarized in Appendix A, and significant differences are depicted in Figure 2.

The figures comprise only those analyzed characteristics for which the effect of treatment and the effect of disease duration *F*-value was > |9|, *p* < 1 × 10^−4^, according to the linear mixed-effects models.

#### 3.2.3. Levels of Acylcarnitines and Their Ratios during the 5.1 Years of Disease Duration and Antipsychotic Treatment

Similar to previous results, the present analysis showed changes in the profiles of measured LCAC and SCAC levels when comparing AP-naïve patients and CSs. Of LCASs: C16 (*t*_(73)_ = 5.25, *p* < 1 × 10^−4^), C16:1 (*t*_(71)_ = 3.96, *p* = 2 × 10^−4^), C16:2-OH (*t*_(74)_ = 3.70, *p* = 4 × 10^−4^), C18:1 (*t*_(72)_ = 4.72, *p* < 1 × 10^−4^), C18:1-OH (*t*_(74)_ = 3.27, *p* = 2 × 10^−3^), C18:2 (*t*_(72)_ = 3.87, *p* = 2 × 10^−4^), and SCACs: C3-DC(C4-OH) (*t*_(74)_ = 4.28, *p* = 1 × 10^−4^), C5-OH(C3-DC-M) (*t*_(74)_ = 4.94, *p* < 1 × 10^−4^), C4:1 (*t*_(74)_ = 3.40, *p* = 1 × 10^−3^), C5-M-DC (*t*_(74)_ = 3.23, *p* = 2 × 10^−3^), C5:1 (*t*_(74)_ = 4.06, *p* < 1 × 10^−4^), and C5:1-DC (*t*_(74)_ = 3.53, *p* = 7 × 10^−4^) were upregulated and C3 (*t*_(73)_ = -5.79, *p* < 1 × 10^−4^) as well al C4 (*t*_(73)_ = −5.64, *p* < 1 × 10^−4^) were downregulated.

After 0.6 years of AP treatment, alterations in the AC profile reduced. However, for the aforementioned ACs, similar profile changes persisted for the following biomolecules: C16:2-OH, C5-OH(C3-DC-M), C4, C4:1, C5:1, C5-M-DC, and C5:1-DC.

During the 5.1years follow-up, most serum levels of LCACs [C16 (*t*_(73)_ = 3.02, *p* = 3 × 10^−3^), C16-OH (*t*_(74)_ = 3.26, *p* = 2 × 10^−3^), C16:1 (*t*_(71)_ = 3.40, *p* = 1 × 10^−3^), C16:1-OH (*t*_(74)_ = 3.56, *p* = 6 × 10^−4^), C16:2 (*t*_(71)_ = 4.26, *p* = 1 × 10^−4^), C16:2-OH (*t*_(74)_ = 2.97, *p* = 4 × 10^−3^), C18 (*t*_(74)_ = 3.04, *p* = 3 × 10^−3^), C18:1 (*t*_(72)_ = 4.01, *p* = 1 × 10^−4^), C18:1-OH (*t*_(74)_ = 3.48, *p* = 9 × 10^−4^), C18:2 (*t*_(72)_ = 4.33, *p* < 1 × 10^−4^)], atcetylcarnitine (C2) (*t*_(73)_ = 4.24, *p* = 1 × 10^−4^), and SCACs: [C3 (*t*_(73)_ = 5.27, *p* < 1 × 10^−4^), C5-OH(C3-DC-M) (*t*_(74)_ = 4.69, *p* < 1 × 10^−4^), C3:1 (*t*_(74)_ = 3.49, *p* = 1 × 10^−3^), C4 (*t*_(73)_ = 4.72, *p* < 1 × 10^−4^), C4:1 (*t*_(74)_ = 5.23, *p* < 1 × 10^−4^), C5:1 (*t*_(74)_ = 4.46, *p* < 1 × 10^−4^), C5:1-DC (*t*_(74)_ = 3.14, *p* = 2 × 10^−3^)], and hexoses (*t*_(73)_ = 3.40, *p* = 1 × 10^−3^) were elevated. A significant increase in the level of hexoses by the fifth year indicates a glucose metabolism disorder.

Based on these findings, we calculated indices or ratios of ACs and ratios of ACs to carnitine. According to the results, values of the indices of (C16 + C18)/C0 (*t*_(73)_ = 3.43, *p* = 1 × 10^−3^) and (C16 + C18:1)/C2 (*t*_(71)_ = 4.18, *p* = 1 × 10^−4^) were elevated among drug-naïve patients compared to CSs. During the follow-up period, the index value of (C16 + C18:1)/C2 showed a decreasing direction, which, however, did not exceed the conservative statistical significance threshold set by us. However, the value of the index of (C2 + C3 + C4 + C5)/C0 at the time of enrollment of FEP patients was reduced compared to CSs, and 5.1 years of continuation of disease and AP treatment resulted in an upward trend (*t*_(68)_ = 3.00, *p* = 4 × 10^−3^) in the index value.

The results are detailed in Appendix A and significant differences are shown in Figure 3 (including LCACs: C16, C16:2, C18:1, C18:2; SCACs: C2, C3, C4, C4:1, C5-OH(C3-DC-M), and ratios of (C16 + C18:1)/C2, and (C2 + C3 + C4 + C5)/C0).

## 4. Discussion

Classically pathophysiologic theories of chronic psychotic disorders have been linked to disrupted brain functioning. However, accumulating evidence suggests that the systemic inflammation and disturbances in metabolism seen in SSD may constitute an important underlying cause and progression of the disorder. It has been recently argued by Prestwood and colleagues [42] that the pathogenesis of SSD leads to compensatory immunological and metabolic changes that represent the body’s attempt to restore homeostasis and some of the alterations in the immune system that are intrinsic to SSD are normalized by APs, while others are not. At the same time, APs generally exacerbate the metabolic disturbances that are characteristic of SSD. Although the exact relationships between the psychopathological symptoms and inflammatory and metabolic disturbances still need further investigation, it is clear that alterations within these domains are interlinked.

To the best of our knowledge, our present study is the first to address the dynamic and simultaneous changes in the serum levels of inflammatory, MetS-related proteomic markers and ACs over five years following the onset of FEP. These alterations appeared against the background of persistent BMI and waist circumference increases in the sample of patients. Meta-analyses [1,43] and our previous studies [31,32] have convincingly shown that patients with a chronic psychotic disorder have an increased risk of abdominal obesity, diabetes and MetS when compared with cohort-matched general population controls. It has been suggested that waist circumference appears to capture a major part of the health risks related to the amount of body fat [44]. Although factors contributing to the increased prevalence of unwanted metabolic events in SSD include the use of AP drugs as well as a range of behavioural components, including an unhealthy lifestyle associated with low physical activity, poor dietary habits and smoking, it has convincingly been shown that there are underlying inflammatory processes that could contribute to the metabolic dysregulation [45].

Chronically activated macrophages and a cluster of differentiation CD4 T helper (Th) cells secrete a major portion of the cytokines, which play important roles in the communication between cells in the immune system, as they have stimulatory or inhibitory effects, and their role may change depending on context [46]. When the CD4 Th cells are activated, they can differentiate into Th1 and Th2 effector cells, producing different types of cytokines. Th1 cells produce IFN-γ, TNF-α, IL-2, which have strong pro-inflammatory properties, while the Th2 cells stimulation produces IL-1, IL-3, IL-4, IL-5, IL-6, and IL-10 with mixed effects; however, these mainly counteract Th1 action. In addition to the Th1 and Th2 cytokines, there is evidence that Treg exert its effect primarily by secreting IL-10 and transforming growth factor beta, which suppress the inflammatory response of the aforementioned cytokines and represent the T-Regulatory response [3]. Previous research has concluded that the balance of pro-inflammatory and anti-inflammatory cytokines is concomitantly disrupted in SSD. However, subsequent studies have repeatedly demonstrated that cytokine changes do not clearly fit into in an imbalance between the Th1 and Th2 arms of the immune system [45]. Rather, it is hypothesized that there is an enhanced pro-inflammatory response instead of clear-cut Th1 or Th2 activity [47]. In addition, serum ferritin is widely recognized as a marker of acute and chronic inflammation, and is nonspecifically elevated in a wide range of inflammatory conditions [48]. Serum ferritin is an important inflammatory disease marker, as it is mainly a leakage product from damaged cells [49].

Several studies, including meta-analyses, have confirmed the presence of inflammatory and immune processes dysregulations in the acute and chronic phases of the disease [3,4,6,50], although not in all SSD patients [51,52,53]. Uptegrove et al. [6], in a meta-analysis, included 570 AP-naïve FEP patients compared to 683 CSs and identified that IL-1B, IL-6, and TNF-α were significantly elevated among the patient group and there was no significant difference in IL-2, IL-4, and IFN-γ between groups. According to a recently published meta-analysis by Dunleavy et al. [3], a significant elevation was reported for IFN-γ, and IL-6 in 651 AP-naïve FEP patients compared to 521 CSs and non-significant change was reported for TNF-α, IL-1B, IL-2, IL-4, IL-8 and IL-10. In those studies that included chronic patients, there are also discrepancies. Potvin et al. [47] included 2298 patients with SSD and 1858 healthy volunteers and assessed cytokines (IFN-γ, IL-2, IL-4, IL-1B, TNF-α, IL-6, IL-10, among others), and the results showed that an increase occurred in IL-6 and a decrease occurred in IL-2 in SSD, and no significant differences were obtained for IFN-γ, IL-4, IL-1B, TNF-α, and IL-10. A meta-analysis by Goldsmith et al. [50] summarized the existing findings regarding cytokine alterations in SSD and also compared results from acute and chronic phases. IL-6 and TNF-α were elevated in the acute phase of SSD. After treatment, IL-6 decreased, but TNF-α did not change. In the chronic state, IL-6 and IL-1B were elevated in the SSD group. Furthermore, it has been suggested that IL-6, IL-2, and probably IL-1β, can be considered as state markers in SSD, which decrease after the initiation of AP therapy, whereas TNF-α and IFN-γ have been considered as trait markers [54,55]. Earlier, Kim and co-workers [56] reported that effective AP treatment in FEP patients increased the serum level of IL-4 and reduced the level of IFN-γ. Furthermore, it has been argued that typical and atypical AP drugs may modulate the production of cytokines or cytokine receptors differently. An experimental study by Sugino et al. [57] demonstrated that some atypical APs can up-regulate IL-10. Additionally, there are also suggestions that changes in cytokine levels occur independently of AP treatment [55].

In this regard, the added materials in our study (Figure 2; Appendix A) illustrate the dynamics of all inflammatory markers, comparing CSs, AP-naïve FEP patients, FEP patients after 0.6 years and after 5.1 years of AP treatment. However, it is necessary to underline several important outcomes. Based on a conservative significance level, our results revealed that AP-naïve FEP patients had, in comparison to CSs, higher levels of IL-2, IL-4, IL-6, and IL-1A, and ferritin. After 0.6 years of AP treatment, only the serum level of IL-4 was still elevated. Against the background of further AP treatment for 5.1 years, the previously manifested positive effect on the profile of inflammatory markers abated, and an increase in serum levels of IL-2, IL-6, IL-1A, IFN-γ, IL-4, and IL-10 appeared, emphasizing the profound return of inflammatory imbalance as the disease became chronic. In addition, the substantial elevation of ferritin level indicates intensified cell damage. Thus, the current study fully agrees with the suggestion that AP treatment probably inflicts a mixture of favorable and unfavorable changes on immunologic abnormalities in SSD that may ultimately promote chronic low-grade inflammation with repeated use [42]. However, discrepancies in the research findings may result from the consideration patients with chronic psychotic disorders as a single group, although a low-grade inflammatory process has been shown to characterize a subset of patients [52]. Additionally, alterations in cytokine levels may be related to at least four different patient categories, including AP-naïve FEP patients, non-drug-naïve FEP patients, stable chronic, and chronic patients in acute relapse. Furthermore, disease duration, the severity and nature of symptoms and the administration of AP-drugs are correlated with the levels of certain cytokines [3,6,50,58]. Despite the emerging complexity, the continued need to study the nature of the inflammatory process in SSD, including at different stages of the disease, has been increasingly emphasized, and this has been confirmed by the continued increase in the number of publications on this topic in recent decades [59].

Furthermore, activation of the immune system in SSD occurs not only in the periphery but also in the brain. Activated macrophages and Th1/Th2 cells release cytokines that may increase the permeability of the blood–brain barrier and cause the overactivation of astrocytes and microglia [60], and these cytokines bind to specific receptors on neurons and adversely affect neurotransmitters [52,61]. They are also known to affect kynurenine pathway regulation [62].

Moreover, the peripheral and central changes are not only connected through immune regulators but also via the endocrine and metabolic systems. Studies have convincingly shown that metabolic abnormalities precede the psychotic disorder [63]; they worsen rapidly after the diagnosis of FEP [64,65], and deteriorate even more in the chronic phase of the disease, against the background of AP treatment [4,11]. The elevation of hexoses at the timepoint of 5.1 years indicates the appearance of disturbances in glucose metabolism. The indispensable link between the adipose tissue, metabolism, and the immune system is adipokines and their regulation. The appropriate secretion of adipokines, including the peptide hormone leptin, plays a crucial role in the ability of adipose tissue to rapidly alternate between the storage and mobilization of fat [66]. Intriguingly, leptin not only regulates food intake, body mass and glucose homeostasis, but also acts as a pro-inflammatory cytokine [67]. Meta-analyses have shown that there is an increase in leptin levels induced by APs across multiple psychiatric populations, including in the group of SSD patients compared to CSs [68,69], but it has also been shown that hyperleptinemia in SSD is not unequivocally related to the consequences of APs [70]. The results of our study are in the same direction. In the patient group, a significant increase in leptin levels occurred in the fifth year.

To ensure effective metabolic control, the sufficient presence of adipose tissue, where energy is stored as triglycerides, is necessary. By regulating the size and number of adipocytes, the structure of adipose tissue is reorganized, and immune cells also contribute to this process. Through the functions of M2 macrophage cells of the innate immune system, which are mainly found in visceral adipose tissue, the anti-inflammatory and insulin-sensitizing effects are mainly manifested in lean people. In contrast, in obese individuals, relatively more pro-inflammatory effects are exerted by these immune cells or a higher number of M1 macrophages, which also results in greater insulin resistance [71]. However, it has been observed that the approach based on the M1/M2 change is an oversimplification and that these two polarizations represent the extremes of the spectrum of activated macrophage states [72]. Free FAs that are elevated due to diet or obesity may also promote inflammation by binding to Toll-like receptor (TLR) 2 and 4, resulting in activation of the C-jun N-terminal kinase (Junk) and NF-*κ*B inflammatory pathways [73]. Adipose tissue accumulation promotes ectopic lipid deposition in other tissues, such as the liver and muscle, where it has a lipotoxic effect. Furthermore, lipids can be dispersed intercellularly and impair organ function via paracrine effects, or accumulate intracellularly and be associated with decreased insulin sensitivity [74]. Either way, metabolic overload causes increased oxidative stress, which contributes to cellular dysfunction [75]. The liver is the central organ that controls lipid homeostasis by means of complexly regulated biochemical, signalling and cellular pathways. For FAs metabolism, mitochondrial β-oxidation is the primary route for the oxidation of the majority of FA found in hepatocytes, including short- (<C4), medium- (C4–C12), and long-chain (C12–C20) FA [76].

Different ACs have different roles in physiological and pathophysiological conditions. Acetylcarnitine (C2) derives from carbohydrate catabolism and from the terminal product of the β-oxidation. The concentration of C2 represents an important mechanism for buffering the metabolic status between fed (glucose oxidation) and fasted (fat oxidation) states, referred to as metabolic flexibility. Under conditions of the continuous oversupply of food and heightened substrate, competition mitochondria lose their capacity to switch freely between alternative forms of carbon energy, and metabolic health deteriorates [77]. SCACs (including C3, C4-OH, C5, C5-OH) can be derived from branched-chain amino acids (BCAA), which have been related to obesity through several potential mechanisms [15,78,79]. The accumulation of LCASs reflects the impaired FA oxidation that has been linked with insulin resistance and oxidative stress caused by mitochondrial overload [80].

Previous publications by us [30] and others [81,82] have shown that changes in the profile of ACs are important in the manifestation of the FEP, in describing the AP treatment effect, as well as in characterizing the chronic phase of SSD. The present study highlights that, among many ACs, the levels of SCACs: C4-OH, and C5-OH and levels of LCACs: C16, and C18:1 were clearly increased and contrary, SCACs: C3, and C4 were significantly decreased, while the AP-naïve patients were compared to CSs. Against the background of 0.6 years of AP treatment, a significant decrease in the serum level of C4 and an increase in C5-OH persisted. In addition, increases in C4:1, and C16:2-OH, which, before treatment, showed an upward trend in the same direction, appeared. Five years of AP treatment caused a clear elevation of SCACs (C2, C3, C4, C4:1, C5:1, C5-OH, C5:1-DC) and LCACs (C16, C16-OH, C16:1, C16:1-OH, C16:2, C16:2-OH, C18, C18:1, C18:1-0H, C18:2). Cao et al. [81] demonstrated that the individuals with schizophrenia showed significantly higher levels of C4-OH and C16:1, but a lower concentrations of C3 when compared with healthy controls and the increase in C3 and C4 emerged after AP-treatment. Furthermore, our recent studies (on the same sample) showed that 5.1 years of continuous AP treatment causes shifts in the general metabolic profile (including amino acids, biogenic amines, and expanded endocannabinoid system) of patients [31,32]. Taking into account the changes in low-grade inflammation, oxidative stress and metabolic changes allowed for us to hypothesize that LCACs likely contribute to the signs of low-grade inflammation and oxidative stress, whereas SCACs are implicated in the regulation of energy metabolism (including a ketogenetic shift in metabolism of BCAA) involving also the development of MetS. The data highlighted in the present study favour this conclusion because long-term AP treatment is accompanied by clear-cut pro-inflammatory changes and disturbances in glucose utilization, together with shifts in lipid metabolism (including ketogenesis) and in energy utilization.

The complexity of the pathogenetic mechanisms of the existence or development of dysregulation in inflammatory and metabolic pathways in SSD at least partially determines its heterogeneity and insufficient treatment effectiveness. Several previously described molecular mechanisms related to metabolic inflexibility in SSD are modulated by PPARs. These ligand-activated transcription factors are involved in the regulation of numerous biological processes, including lipid synthesis and oxidation, adipocyte differentiation, glucose metabolism and insulin sensitivity, (obesity-induced) inflammation and the expression of immunoregulatory genes [83,84].

PPAR*α* can effectively induce the expression of numerous genes involved in a plethora of lipid metabolic pathways, including microsomal, peroxisomal, and mitochondrial FA β-oxidation, thus yielding substrates for ketone body synthesis to provide energy for peripheral tissues, FA elongation and desaturation, synthesis and breakdown of TGs and lipid droplets, and various other metabolic pathways and genes [85]. PPARγ is a crucial transcriptional regulator of adipogenesis [86]. In addition, PPARγ promotes the cells’ sensitivity to insulin [87]. Based on the effect on the activity of lipid synthesis, PPARγ and cannabinoid CB1 receptors can be considered antagonistic. Our recent study [32] showed an increase in the levels of the endogenous agonist of cannabinoid CB1 receptors 2-arachidonyl glycerol and its potential precursor molecules 5.1 years after the start of treatment. At the same time, we observed a significant reduction in the PPARγ agonist linoleoyl ethanolamide and its precursor molecules [32]. These changes can be taken as a significant shift favouring the enhancement of lipid synthesis in patients 5.1 years after starting treatment.

Regarding PPAR*β*/*δ*, studies indicate a protective role of this PPAR isoform in the control of lipid metabolism and the development of inflammatory processes [88]. Although PPARs affect inflammation through direct and indirect mechanisms, the anti-inflammatory properties of PPARs are mainly achieved by inhibiting NF-*κ*B, which is the pro-inflammatory nuclear transcription factor (for more details, see th review provided by Wahli and Michalik [84]). Furthermore, the expression of all PPAR isoforms has been described in the central nervous system, but PPARα and PPARγ have received more attention, primarily due to the modulatory effect of glutamate neurotransmission, neuroprotective, and anti-inflammatory properties [89]. Therefore PPARs are considered as potential therapeutic targets for several neuropathological conditions [90]. Regarding SSD, Chase et al. [91] reported a significant increase in PPARγ and a decrease in PPARα mRNA levels, with additional BMI interactions in patients with schizophrenia. Costa et al. [92] demonstrated that a decrease in PPARα expression is associated with neuroinflammation and susceptibility to chronic psychotic disorder.

The current study is not free of limitations. First, the sample sizes are relatively small, although large enough to enable the detection of changes in the levels of biomolecules [93]. Second, we collected data from CSs at one point in time. Third, based on the naturalistic study design, the choice, or the dosage regimen of AP after enrolment and during the 5-year follow-up period did not have any description. Therefore, we could not demonstrate a direct therapeutic dose impact of the APs on the levels of biomolecules. Furthermore, patients used APs to control psychotic symptoms during the follow-up visits. However, other psychotropic medications may have been added to their regimen, which may also affect the levels of metabolic and inflammatory markers. Because the antidepressants and mood stabilizers were only added to the treatment regimen for patients with accompanying mood swings, the use of these drugs was not stable over time. Therefore, their effect on biomarkers was not systematically taken into account when performing the analyses. Fourth, we described that some patients and CSs had used cannabis during their lifetime. We were not able to ascertain the potency of the cannabis consumed, and the frequency of use based on the subjects’ memories. Therefore, we did not include this information as a confounding factor in the analyses. Fifth, due to sample sizes, we were unable to generate subgroups of patients based on their level of inflammatory or metabolic status indicators. Fifth, we used commercial immunoassays and targeted metabolomics assay platforms for biomolecule measurement. This influenced the choice of studied markers and did not specifically target all key regulators of immune and metabolic functions.

## 5. Conclusions

At present, no one can replace AP treatment in FEP patients, but the serious side effects of medication should be taken into account, and ways to alleviate these side effects should be considered. Adipose tissue accumulation during the disease progression and AP treatment affects several crucial homeostatic factors via disruption of the mechanisms controlling lipid and carbohydrate metabolism and causes unwanted alterations in cytokine and adipokine profiles. Measurement of biomarkers (including IL-2, IL-4, IL-6, IFN-γ, leptin, selected SCACs and LCACs, and endocannabinoids) at specific time points provides information on the inflammatory and metabolic status of FEP patients. Based on this knowledge, it is possible to intervene to prevent or mitigate unwanted side effects associated with the APs usage. Furthermore, the results of the present study confirmed previous studies suggesting that a metabolic–inflammatory imbalance characterizes AP-naïve patients, and short-term AP treatment has a favourable effect on psychotic symptoms and the recovery of metabolic flexibility and the resolution of low-grade inflammation. However, when the disease becomes chronic, against the background of years of AP treatment, metabolic inflexibility and increased pro-inflammatory activity appear. The results provide indirect confirmation that the imbalance in the PPARs system, which integrates both energy metabolism and immune system functioning at the cellular level, worsens over time among patients with SSD.

## Figures and Tables

**Figure 1 metabolites-12-00983-f001:**
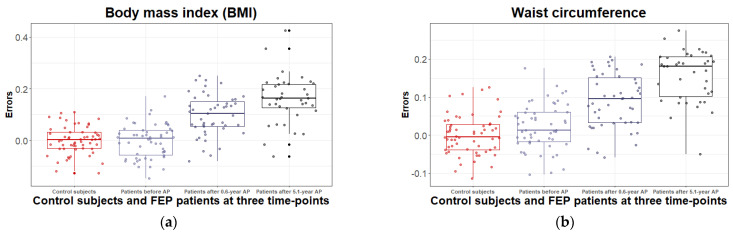
Boxplots of the variation in the errors of BMI (**a**) and waist circumference (**b**) (derived by regressing out covariate effects) for control subjects and first episode (FEP) patients at baseline (before treatment with antipsychotics (AP)), after 0.6 years, and after 5.1-year treatment with AP. The solid horizontal line in each box represents the median. The area above and below the line represents the 50th to the 75th and the 25th to the 50th percentiles, respectively. The whiskers extend to the highest and lowest values contained within 1.5 times the interquartile range of the data. Each calculated error variance is represented as a dot.

**Figure 2 metabolites-12-00983-f002:**
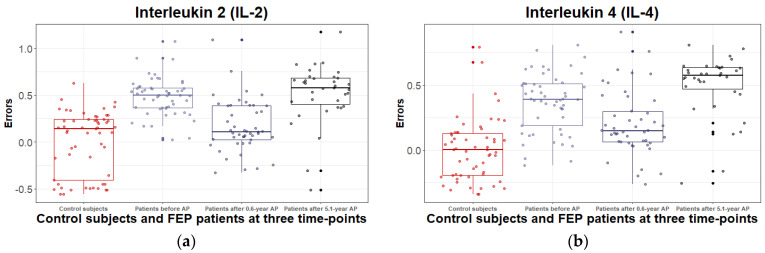
Boxplots of the variation in the errors of log-transformed levels of inflammatory (**a**–**e**) and metabolic (**f**) biomarkers (derived by regressing out covariate effects) for control subjects and first episode (FEP) patients at baseline (before treatment with antipsychotics (AP)), after 0.6 years, and after 5.1-year treatment with AP. The solid horizontal line in each box represents the median. The area above and below the line represents the 50th to the 75th and the 25th to the 50th percentiles, respectively. The whiskers extend to the highest and lowest values contained within 1.5 times the interquartile range of the data. Each calculated error variance is represented as a dot.

**Figure 3 metabolites-12-00983-f003:**
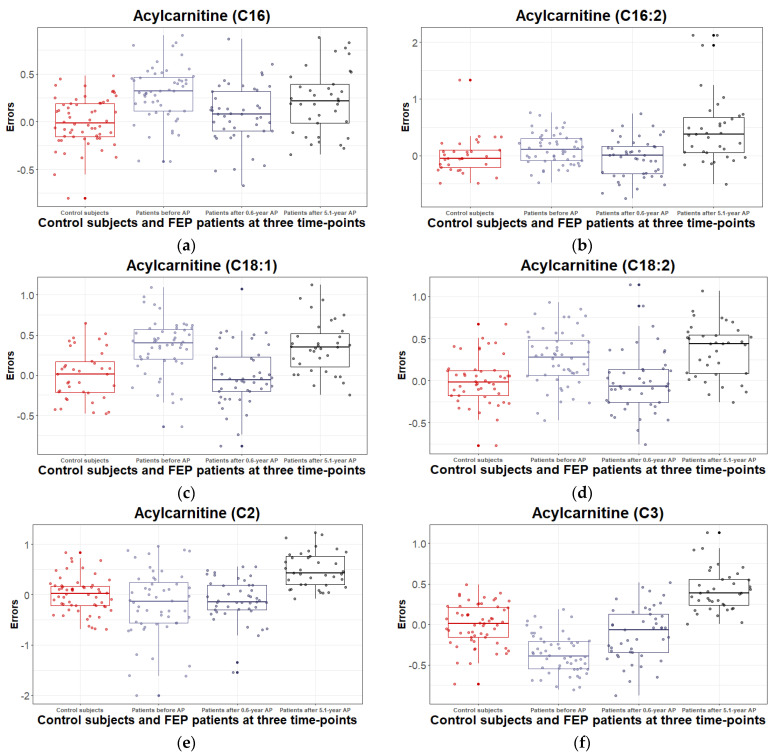
Boxplots of the variation in the errors of log-transformed levels of acylcarnitines (**a**–**i**) and their ratios (**j**,**k**) (derived by regressing out covariate effects) for control subjects and first episode (FEP) patients at baseline (before treatment with antipsychotics (AP)), after 0.6 years, and after 5.1-year treatment with AP. The solid horizontal line in each box represents the median. The area above and below the line represents the 50th to the 75th and the 25th to the 50th percentiles, respectively. The whiskers extend to the highest and lowest values contained within 1.5 times the interquartile range of the data. Each calculated error variance is represented as a dot.

**Table 1 metabolites-12-00983-t001:** Characteristics of control subjects (CSs) and first-episode psychosis (FEP) patients at baseline (before treatment with antipsychotics (FEP_b_), after 0.6-year treatment (FEP_0.6-year_), and after 5.1-year treatment (FEP_5.1-year_) with antipsychotics.

Characteristics	Participants				Comparison between FEP_b_ and CSs
	CSs	FEP_b_	FEP_(0.6-year)_	FEP_(5.1-year)_	FEP_b_ and CSs
Participants	58	54	47	38	
Age (years), mean ± SD (range)	24.7 ± 4.5 (19.1–39.3)	26.6 ± 6.1 (18.7–41.1)	27.3 ± 6.4 (19.3–41.7)	31.8 ± 5.9 (23.7–46.2)	*t*_(110)_ = 1.87, ns
Men (%)	24 (44%)	31 (57%)	27 (51%)	23 (43%)	χ^2^_(1)_ = 3.58, ns
Current cigarette smokers (*n*%)	15 (26%)	18 (33%)	16 (30%)	20 (37%)	χ^2^_(1)_ = 0.33, ns
BMI (kg/m^2^) mean ± SD	22.6 ± 2.8	22.8 ± 3.0	25.3 ± 3.9 ^a^	27.8 ± 4.7 ^b,c^	*t*_(110)_ = 0.34, ns
Waist circum- ference (cm) mean ± SD	77.9 ± 12.6	81.0 ± 9.89	86.3 ± 11.2 ^a^	94.4 ± 11.2 ^b,c^	*t*_(110)_ = −1.68, ns
BPRS score mean ± SD (range)	-	49.9 ± 15.4 (13–85)	22.9 ± 12.7 ^a^ (2–48)	14.2 ± 10.5 ^b,c^ (0–49)	-

BMI—body mass index; BPRS—Brief Psychiatric Rating Scale. ^a^—Statistically significant difference (*p* < 0.05) between patients before (FEP_b_) and after 0.6-year treatment (FEP_(0.6-year)_). ^b^—Statistically significant difference (*p* < 0.05) between 0.6-year (FEP_(0.6-year)_) and 5.1-year treatment (FEP_(5.1-year)_)_._
^c^—Statistically significant difference (*p* < 0.05) between patients before (FEP_b_) and after 5.1-year treatment (FEP_(5.1-year)_). ns: not significant (*p* ≥ 0.05).

## Data Availability

The data that support the findings of this study are available on request from the co-author L.H. (liina.haring@kliinikum.ee). The data are not publicly available due to possibility of participants’ identification.

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
