# Peer review of "A Marked Low-Grade Inflammation and a Significant Deterioration in Metabolic Status in First-Episode Schizophrenia: A Five-Year Follow-Up Study"

_metabolites, 2022, doi:10.3390/metabo12100983_

Round 1

Reviewer 1 Report

Hello Dears ;

This is a good Research.

Please explain more about Benzodiazepine administration the night before data collection.

A more complete explanation should be given about the exclude criteria.

Reviewer 2 Report

Many thanks to the editor and authors for the opportunity to review this manuscript. Ia a very interesting and original investigation that studies the dynamic and simultaneous changes in serum levels of inflammatory, MetS-related proteomic markers and ACs  over five years following the onset of FEP.

However, I would propose some minor changes prior to its publication. I provide all details in comments below:

-          In the first place, I would appreciate it being detailed in the description of the methodology, if patients suffering from an acute pathology (for example, infection, fever, allergy) or taking any drug (for example, anti-inflammatories) or vaccines that could alter their inflammatory system were excluded.

-          Patients taking other psychotropic drugs that may affect the metabolic or inflammatory system were included (antidepressant and mood stabilizers).

-          Moreover, patients with a history or consumption of toxic substances were also included. These factors were taken into account when performing the statistical analyses.
